# Clinical Characteristics and Outcomes of *S. Aureus* Bacteremia in Patients Receiving Total Parenteral Nutrition

**DOI:** 10.3390/nu12103131

**Published:** 2020-10-14

**Authors:** Michelle Gompelman, Renée A. M. Tuinte, Marvin A. H. Berrevoets, Chantal P. Bleeker-Rovers, Geert J. A. Wanten

**Affiliations:** 1Intestinal Failure Unit, Department of Gastroenterology and Hepatology, Radboud University Medical Center, 6500 HB Nijmegen, The Netherlands; renee.tuinte@radboudumc.nl (R.A.M.T.); geert.wanten@radboudumc.nl (G.J.A.W.); 2Department of Internal Medicine, Division of Infectious Diseases, Radboud university medical center, 6500 HB Nijmegen, The Netherlands; m.berrevoets@etz.nl (M.A.H.B.); chantal.bleeker-rovers@radboudumc.nl (C.P.B.-R.)

**Keywords:** total parenteral nutrition, *S. aureus* bacteremia, catheter-related bloodstream infection

## Abstract

*Background:* Patients on total parenteral nutrition (TPN) are at risk of developing central line-associated infections. Specifically, *Staphylococcus aureus* bacteremia (SAB) is feared for its high complication rates. This prospective cohort study compares characteristics, clinical course and outcome of SAB in patients with and without TPN support. *Methods:* Clinical and microbiological data from all patients with positive blood cultures for *S. aureus* from two facilities, including our referral center for TPN support, were retrieved (period 2013–2020). Primary outcome was overall mortality, and included survival analysis using a multivariate Cox regression model. Secondary outcomes comprised a comparison of clinical characteristics and outcomes between both patient groups and analysis of factors associated with complicated outcome (e.g., endocarditis, deep-seated foci, relapse and death) in patients on TPN specifically. *Results:* A total of 620 SAB cases were analyzed, of which 53 cases received TPN at the moment the blood culture was taken. Patients in the TPN group were more frequently female, younger and had less comorbidity (*p* < 0.001). In-hospital death and overall mortality were significantly lower in TPN patients (4% vs. 18%, *p* = 0.004 and 10% vs. 34%, *p* < 0.001, respectively). Positive follow-up blood cultures, delayed onset of therapy and previous catheter problems were associated with a higher incidence of complicated SAB outcome in patients on TPN. *Conclusion:* Our data show that patients on TPN have a milder course of SAB with lower mortality rates compared to non-TPN SAB patients.

## 1. Introduction

Central line-associated bloodstream infections (CLABSIs) are the most common and serious complications for patients with intestinal failure receiving total parenteral nutrition (TPN) [1,2,3]. The overall incidence of CLABSIs in patients receiving TPN ranges from 0.38 to 4.58 episodes per 1000 catheter days and account for nearly 70% of all hospital admissions [4,5,6]. *Staphylococcus aureus* is the causative pathogen in approximately 10–26% of the cases [4,7,8,9].

*S. aureus* can cause a broad range of community-acquired, hospital-acquired and/or healthcare-associated infections [10]. Among these, *S. aureus* bacteremia (SAB) presents as a life-threatening infection that is associated with an all-cause mortality ranging from 20 to 30% in the general population [11,12,13,14]. The most common co-existent factors for SAB are the presence of a peripheral venous catheter or central venous catheter (CVC) [15]. Treatment failure is present in up to 50% of the *S. aureus* CLABSIs and is often complicated by endovascular infection foci like endocarditis or suppurative thrombophlebitis [16]. As a result, *S. aureus* CLABSIs frequently lead to long hospitalizations, prolonged antibiotic courses and, if recurrent, eventually to loss of vascular access [17,18,19]. Nevertheless, mortality from SAB in patients with a CVC as the main focus seems lower than in patients with other dominant foci [12,13,14,15,16,17,18,19,20], with a 7–21% attributable mortality rate at 30 days [21,22,23].

However, data regarding the patient characteristics and clinical outcomes, including mortality, of CLABSI caused by *S. aureus* in patients on TPN are scarce, and factors associated with complicated outcomes in this group have not been investigated yet. Therefore, the aim of the present study is to map the epidemiologic data and outcomes of SAB in patients receiving TPN, and in comparison to patients without TPN.

## 2. Materials and Methods

### 2.1. Study Design and Setting

Data were analyzed from a prospective cohort study (*Staphylococcus aureus* bacteremia monitor), performed at two centers, including a tertiary referral center for TPN support, located in the Netherlands. We requested all consecutive blood cultures with growth of *S. aureus* from the microbiology laboratory between January 2013 and May 2020. All adult patients (>17 years) with at least one positive blood culture and clinical signs of bacteremia were included. If patients developed another episode of SAB during the study period, this was considered as a new case when it was at least three months after the discontinuation of antibiotic treatment. To increase the number of TPN patients with SAB, we consulted the Intestinal Failure (IF) Registry Nijmegen database [24], which contains data of all TPN patients in the Radboud UMC, including complete data regarding bloodstream infections treated in other hospitals.

### 2.2. Data Collection

We assessed the medical records of all patients and gathered electronic data on patient demographic characteristics. The data were retrieved from the infectious disease (ID) consultation documentation and results of diagnostic studies (including laboratory, microbiologic and imaging data). We determined onset of bacteremia, presence of foreign body material and intravascular catheters, clinical parameters at SAB onset, diagnostic tests, antimicrobial therapy and survival outcomes during and after hospital stay and at 3 and 6 months. Preexisting underlying disease and comorbidity were calculated according to the Charlson comorbidity index (CCI) [25] adjusted for age. Data were entered in a secure database (Castor EDC, Amsterdam, The Netherlands) and processed anonymously for further analysis. Follow up time was 6 months, since the majority of patients (>80%) had regular follow-up appointments at the infectious diseases outpatient clinic during this period.

### 2.3. Ethical Consideration

This study was, according to Dutch law, cleared from the requirement of approval by an ethics committee, because of the observational character of this study and the anonymous processing of data. The regional institutional review board approved this study (2015–2257) [26].

### 2.4. Management of Central Line Sepsis and S. aureus Bacteremia in Patients with TPN Support

Prior to the start of parenteral nutrition administration in the home situation (HPN), patients are trained in aseptic handling of their CVC by specialized nurses during a training period of 1–2 weeks in our referral center. This training is implemented according to a standardized protocol and is in line with recent European Society for Clinical Nutrition and Metabolism (ESPEN) guidelines [27]. Extra emphasis is put on the awareness of potential central line sepsis by instructing the patients to monitor their body temperature regularly and whenever signs of infection are present. In addition, they are strongly advised to contact our referral center directly in case of fever (>37.5 °C) or other signs of infection (e.g., inflammation at the insertion site of the CVC) arise. More details about the CVC care and central line sepsis management protocol can be found elsewhere [28].

All patients with *S. aureus* bacteremia (including patients with TPN) were managed according to the advice of the national antimicrobial stewardship program, which included the performance of echocardiography and consultation by an ID specialist. Patients were treated according to the national guideline for SAB, which is in accordance with the Infectious Diseases Society of America guideline [29]. Yet, our institutional guideline recommends antimicrobial therapy for 2 weeks instead of 4–6 weeks for patients with risk factors for complicated outcome (e.g., presence of CVC), but without endocarditis or signs of metastatic infection on 18-fluor-FDG positron emission tomography/computed tomography (^18^F-FDG PET/CT); these patients were considered as uncomplicated SAB cases [26].

### 2.5. Outcomes and Definitions

The primary outcome was overall mortality in the TPN- and non-TPN groups. Mortality was calculated from the date of first positive blood culture till the date of death or end of follow-up period (6 months). Attributable mortality (death due to SAB) was defined as clinical or microbiological evidence of infection with *S. aureus* at time of death and no other explanation for cause of death. Patients were considered to be cured in the case of the resolution of signs of infection, and no positive follow-up blood cultures were present after the discontinuation of antibiotic treatment. Relapse of SAB was defined as another episode of SAB within 3 months after the end of antibiotic treatment.

SAB outcome was considered complicated when patients had any of the following: infective endocarditis; metastatic infection; non-retainable infected foreign body; relapse of infection [14]. Metastatic infection was identified using ^18^F-FDG-PET/CT scanning or with evidence from other relevant imaging studies. Infective endocarditis was diagnosed according to the modified Dukes criteria [30]. Predictors identified by the literature for complicated SAB were positive follow-up blood cultures, community acquisition, fever ≥72 h and skin abnormalities suggesting active systemic infection [14]. Patients received antibiotic treatment according to our national SAB guideline [31].

All used definitions were according to commonly used guidelines and are listed in the supplementary appendix (Appendix A, Table A1) [22,32]. The mode of acquisition was classified into community acquired, healthcare associated or hospital acquired according to Bishara et al. [33].

All patients who had a CVC that was present for at least 48 h and received parenteral nutrition at the time of positive blood culture were included in the TPN group and, among these patients, most (39/41) consisted of patients who received long-term (home) parenteral nutrition (HPN).

### 2.6. Statistical Methods

For the descriptive statistics, continuous variables were compared by use of the independent samples t-test or Mann–Whitney test when non-parametric testing was indicated. Categoric variables were compared by use of the chi-squared test or Fisher’s exact test when values were small (<5). Differences were considered to be statistically significant at a two-sided *p*-value < 0.05. Sensitivity analysis was done for three specific variables to examine whether these influenced our data and outcomes, namely (1) patients treated in other hospitals with a significant amount of missing data, (2) cases with missing data on primary outcome and (3) short-term parenteral nutrition (e.g., <1 month). The primary endpoint of our analyses was a comparison of survival time between TPN and non-TPN groups. Survival was analyzed using Kaplan–Meier plotters with a log-rank test and multivariate Cox regression analysis. The patient-dependent variables entered in the multivariate analysis were those with at least 90% non-missing observations and a univariate *p*-value of <0.1. Because of the relatively low number of events (death) in our cohort, the inclusion of variables was limited to the ones most significant by univariate analysis. Furthermore, univariate logistic regression analysis was done to identify factors associated with complicated SAB. All statistical analyses were performed using SPSS statistics, version 25.3 (Armonk, NY, USA).

## 3. Results

### 3.1. Comparison of Patient Characteristics

During the study period, a total of 646 cases of *S. aureus* bacteremia were screened for inclusion (Figure 1). Eventually, 620 cases in 604 patients could be included in the analysis. Fifty-three of these cases occurred in 41 patients receiving TPN. Patients receiving TPN were more frequently female (68.3% vs. 37.1%, *p* < 0.001), younger (mean age 53.4 vs. 63.2, *p* < = 0.001), had a lower Charlson comorbidity index score (mean 1.90 vs. 3.64, *p* < 0.001) and foreign body material (excluding CVC) was less frequently present (19.5% vs. 39.3%, *p* = 0.012). Risk factors for endocarditis and immune status did not differ between the two groups (Table 1).

### 3.2. Clinical Characteristics of S. aureus Bacteremia

A CVC was most likely the portal of entry in 91% of the patients on TPN and the mode of acquisition was healthcare associated in 83% and hospital acquired in the remaining 17% (Table 2). In non-TPN patients, skin was the most common portal of entry (34%) and onset was almost as frequent for community acquired (36%) as hospital acquired (37%). MRSA rates did not differ between the two groups, with an overall rate of 2.1%. At the onset of the bacteremia, patients receiving TPN showed a significantly lower level of C-reactive protein with a mean of 74.6 mg/L versus 169 mg/L (*p* < 0.001). Percentage of intensive care admissions did not differ between both groups with an occurrence of 23% in TPN patients vs. 29% in non-TPN patients (*p* = 0.40). On presentation, 85% of all patients experienced fever, which lasted for more than 72 h in 32% of all patients. Among patients on TPN, the percentage of persistent fever was remarkably lower (18% vs. 33%, *p* = 0.04). Although deep-seated foci were found with ^18^F-FDG PET-CT in more than half of the TPN cases, this was significantly less than in non-TPN patients (54% vs. 73%, *p* = 0.02), with pulmonary and endovascular foci being the most prevalent (28% and 19%, respectively). In total, there were 20 patients with metastatic infection(s). In 80% of the cases, these were diagnosed with ^18^F-FDG PET/CT scanning.

Half of the TPN patients had signs of inflammation at the CVC insertion site prior to the SAB episode; most frequently, this was redness and tenderness (both 36%), followed by induration (16%) (Figure 2). In 44% (11/25 cases), an exit-site culture was performed shortly before the SAB episode, which was positive for *S. aureus* in 82% of the cases. Nasal *S. aureus* carriage was tested in 30 TPN cases and was positive in the majority of the patients (70%). The CVC was removed according to protocol in all but one TPN case, with a mean of 1.8 (SD 1.7) days from positive blood culture to removal.

### 3.3. Antibiotic Treatment

Significantly more patients on TPN reported an allergy to antibiotics (26.4% vs. 12.0%, *p* = 0.003) (Table 1), and fewer TPN patients were switched to oral antibiotics during treatment (10% vs. 34%, *p* < 0.001) (Table 2). TPN patients less frequently received rifampicin in addition to conventional therapy and fewer patients underwent surgical drainage. No other significant differences in provided treatment were seen.

### 3.4. Mortality and Multivariate Survival Analyses

Follow up data on mortality were available for 604 (of 620) SAB cases. Mortality was substantially lower in patients on TPN at all time periods (Figure 3 and Table 3). Only two (4%) in-hospital deaths occurred in patients on TPN compared to 101 (18%) in non-TPN patients (*p* = 0.004). Forty percent (2/5) of the deaths in TPN patients occurred after hospital discharge in comparison to 43% (75/176) in those not receiving TPN. To analyze if TPN was associated with an increased survival probability, a multivariate Cox regression model was designed, including the significant (*p* < 0.1) variables from univariate analysis (CCI score adjusted for age and risk factors for endocarditis) (Figure 4 and Table 4). Age was excluded from multivariate analysis since a strong correlation with CCI score was present (Pearson R = 0.65, *p* < 0.001). According to the Cox model, receiving TPN resulted in a 38% higher survival probability (adjusted HR 0.38, *p* = 0.03). Both CCI score and risk factors for endocarditis were found to be independent risk factors for mortality (HR 1.32, *p* < 0.001 resp. HR 1.74, *p* < 0.01).

### 3.5. Clinical Characteristics of Complicated SAB in Patients Receiving TPN

When comparing the clinical characteristics of patients receiving total parenteral nutrition with and without complicated SAB it appeared that healthcare-associated onset, higher CRP at time of positive blood culture, recent problems with the central venous catheter, treatment onset >24 h and positive follow-up blood cultures were more prevalent in cases with a complicated SAB (Table 5).

## 4. Discussion

To our knowledge, this is the first study presenting a comprehensive overview of patient characteristics and clinical outcomes of *S. aureus* bacteremia in patients receiving total parenteral nutrition in the setting of chronic intestinal failure. In summary, our data show that patients on TPN have a milder course of SAB with lower mortality rates compared to non-TPN patients. Healthcare-associated onset and increased CRP at onset, together with recent catheter problems, positive follow-up blood cultures and delayed initiation of therapy, were more prevalent in the small subset of TPN patients with a complicated outcome.

Surprisingly, overall mortality (10%) was remarkably lower in SAB patients receiving TPN. An explanation for this may be the fact that these patients are well educated about their infection risk and do not hesitate to call and visit the hospital as soon as signs of infection become present [34]. Although onset of antibiotic treatment after the start of symptoms did not significantly differ between TPN and non-TPN patients, delayed treatment onset was associated with a complicated outcome of SAB in patients on TPN (*p* < 0.001). Next, patients with TPN support were found to be younger, more frequently female and with less comorbidity in comparison to non-TPN patients with *S. aureus* bacteremia. These are all factors that are associated with a lower mortality risk in the general population [12,15,20,35]. Nevertheless, receiving TPN was still independently associated with an increased survival probability in the multivariate Cox model. Lastly, our findings of relatively low mortality rates in TPN patients can in part be the successful outcome of a strict adherence to antimicrobial stewardship programs in the Netherlands (e.g., bedside ID consultation and performance of echocardiography in all patients).

Previous studies conducted in oncology, hemodialysis and ICU patients with a *S. aureus* CLABSI all found higher mortality rates (19–24% [16,36], 10–19% [37,38] and 18–53% [39], respectively). Our data on mortality and survival probabilities are, however, difficult to compare with the previous literature, because reported time periods and used mortality definitions differ highly among studies [11,12,14,34,36,37,38,39,40,41,42,43]. For example, Ghanem et al. [36] reported all-cause mortality at 3 months, while Blot et al. [39] reported a 30-day all-cause mortality, and Fowler et al. [14,43] frequently report on SAB attributable mortality. As said, chosen time periods to determine mortality rates differ widely as well: calculation may be based on days counted between death date and either date of positive blood culture or either date of discharge or even date of treatment discontinuation [11,12,14,34,36,37,38,39,40,41,42,43].

Forty-two percent of the TPN patients developed a complicated SAB, which is in accordance with previous research conducted in the general population (36–42%) [16,44]. In contrast, the study performed in oncology patients with a CVC by Zakhem et al. [16] reported a substantially higher complicated SAB rate of 67%. It is important to note that in contrast to our study, persistent fever and bacteremia beyond 72 h were accounted for as complicated SAB as well. We consider these factors as risk factors for a complicated course, not necessarily being a complicated course or always resulting in it. Besides, some studies include SAB attributable mortality as an indicator for complicated SAB [14,43]. We decided to exclude SAB attributable mortality, since we believe this is a rather subjective outcome.

To establish metastatic infection, ^18^F-FDG PET/CT seems to be a valuable technique in TPN patients as well, since 80% (16/20) of the metastatic infections were diagnosed with ^18^F-FDG PET/CT [26,45]. Thus, only in four patients the metastatic infection was diagnosed with other diagnostic modalities. Moreover, the majority (53%) of the TPN patients received less than 4 weeks of treatment for their *S. aureus* bacteremia since metastatic infection was considered to be ruled out safely with the use of ^18^F-FDG PET/CT.

The total number of metastatic infectious foci was lower in TPN patients compared to non-TPN patients, which is in agreement with previous research in patients with a CVC [46]. Nevertheless, endovascular and pulmonary metastatic infections were more prevalent in patients receiving TPN. These are probably directly related to the presence of a catheter since the endovascular metastatic infections included mostly a septic thrombosis near the catheter. Additionally, previous research showed a higher rate of pulmonary septic emboli when (septic) thrombosis was present [18].

Our findings of clinical characteristics of SAB cases associated with complicated outcome are in agreement with previous findings [14,16]. These studies additionally performed multivariate regression analysis to identify independent risk factors for complicated SAB: the strongest predictor Fowler et al. [14] found was a positive follow-up blood culture at 48 to 96 h. The other study, by Zakhem et al. [16], specifically studied SAB in patients with a CVC and found that catheter site inflammation was an independent predictor of complicated course, which was not associated with complicated outcome in our study. One of the other identified predictors by Fowler et al. [14], fever > 72 h, was also not significantly more prevalent in our study, nor in the study of Zakhem et al. [16]. A new finding of our study was the incidence of recent catheter problems (e.g., recent infection, dislocation or thrombosis) being more prevalent in TPN patients with a complicated outcome. Whether the identified clinical characteristics of our study are independent risk factors in patients receiving TPN as well needs to be confirmed in a larger patient cohort, since we decided not to perform multivariate logistic regression analysis with this small sample size.

Several studies found that failure or delay of CVC removal is an independent risk factor for relapse or hematogenous complications of SAB in patients with a CVC [16,19,43,47,48,49]. Since in our study, in almost all TPN patients, the CVC was removed early in their illness course (according to protocol), we cannot confirm nor contradict these findings.

Our study is subject to several limitations: (1) the study was conducted retrospectively, data collection, however, was performed mostly prospectively and almost all patients were diagnosed and treated according to a predefined protocol, leading to low amounts of missing data; (2) analysis had to be done with a rather small sample size of patients receiving TPN. Nevertheless, regarding the rare nature of the conditions leading to dependency on TPN, much higher patient numbers are difficult to gather, unless data are collected as a multicenter or merged into, for example, a meta-analysis with collected individual participant data; (3) some of the TPN patients might be incorrectly considered as uncomplicated SAB since they died early in their illness course, leading to an underestimation of the true incidence of complicated outcome as well. To overcome this, we performed an additional analysis excluding these patients from analysis: no differences in outcomes were seen.

## 5. Conclusions

We conducted the first study of *S. aureus* bacteremia in chronic TPN patients, resulting in an extensive and robust data analysis with >90% complete follow-up data. It provides new insights about its presentation and outcomes in this vulnerable population, who are at continuous risk for developing severe bloodstream infections. We have showed that patients on TPN have a milder course of SAB with lower mortality rates compared to non-TPN SAB patients. Late onset of antibiotic therapy, previous catheter problems, and positive follow-up blood cultures seem to be associated with complicated outcome. Nevertheless, larger, preferably randomized and multicenter clinical trials are needed to further investigate and validate our findings.

## Figures and Tables

**Figure 1 nutrients-12-03131-f001:**
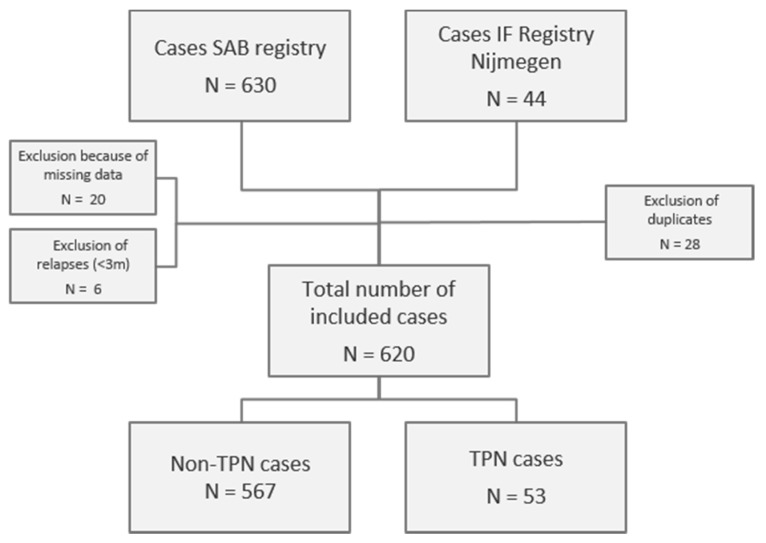
Flowchart of patient acquisition. SAB, *Staphylococcus aureus* bacteremia; TPN, total parenteral nutrition. IF, Intestinal Failure.

**Figure 2 nutrients-12-03131-f002:**
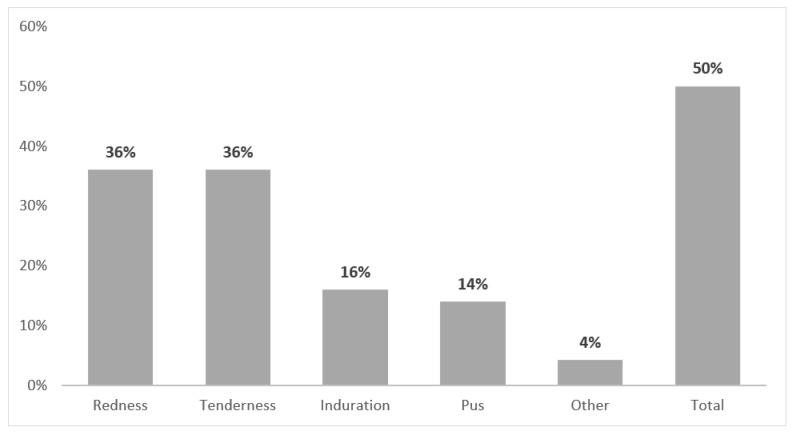
Reported central venous catheter (CVC) insertion abnormalities in patients receiving total parenteral nutrition at time of *S. aureus* bacteremia.

**Figure 3 nutrients-12-03131-f003:**
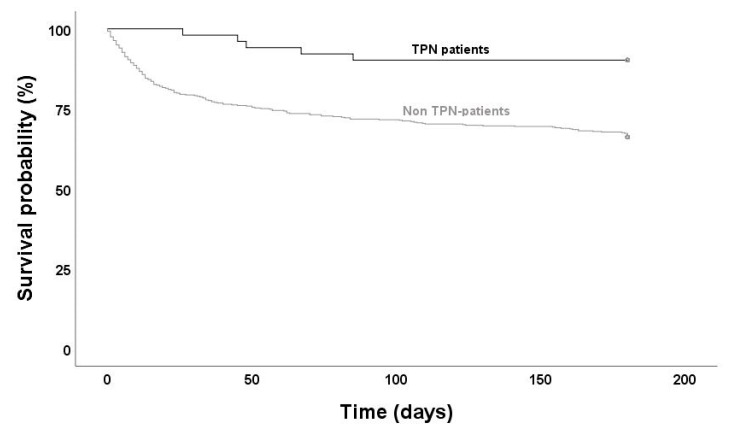
Kaplan–Meier curves for survival time in patients with *S. aureus* bacteremia. TPN: total parenteral nutrition. Log-rank test: *p* = 0.001.

**Figure 4 nutrients-12-03131-f004:**
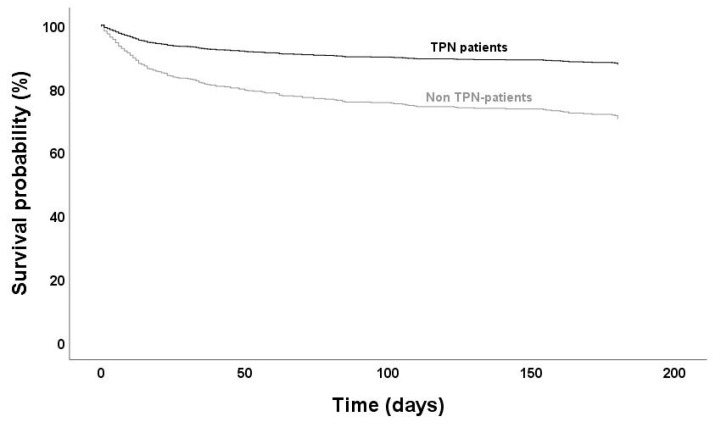
Multivariate Cox regression curves for survival time in patients with *S. aureus* bacteremia. (HR 0.38, CI 0.16–0.93, *p* = 0.03; adjusted for Charlson comorbidity index and presence of risk factors for endocarditis). TPN: total parenteral nutrition.

**Table 1 nutrients-12-03131-t001:** Comparison of patient characteristics.

Patient Characteristics	All Patients *n* = 604	TPN Patients *n* = 41	Non-TPN Patients *n* = 563	*p*-Value
Age, mean (SD)	62.5 (16.7)	53.4 (16.0)	63.2 (16.6)	<0.001
Female (%)	237 (39.2)	28 (68.3)	209 (37.1)	<0.001
Presence of CVC (%)	110 (18.2)	41 (100)	69 (12.2)	<0.001
*Foreign body (%)*	229 (37.9)	8 (19.5)	221 (39.3)	0.012
Joint prosthesis	70 (11.6)	1 (2.4)	69 (12.3)	0.073 *
Pacemaker/ICD	47 (7.8)	2 (4.9)	45 (8.0)	0.761 *
Prosthetic heart valve	52 (8.6)	2 (4.9)	50 (8.9)	0.565 *
Vascular prosthesis	50 (8.3)	4 (9.8)	46 (8.2)	0.766 *
Other ^1^	54 (8.9)	3 (7.3)	51 (9.1)	1.00
Charlson comorbidity index, mean (SD)	3.52 (2.33)	1.90 (1.88)	3.64 (2.32)	<0.001
Immunocompromised ^2^ (%)	139 (23.3)	9 (22.0)	130 (23.4)	0.834
Risk factors for endocarditis ^3^ (%)	63 (10.4)	3 (7.3)	60 (10.7)	0.790 *
Allergy to antibiotics (%)	78 (12.9)	10 (24.4)	68 (12.1)	0.024

**Abbreviations:** CVC: central venous catheter; ICD: implantable cardioverter–defibrillator; SD: standard deviation; TPN: total parenteral nutrition. ^1^ For example, nephrostomy catheter, osteosynthesis material. ^2^ Any condition or usage of immunosuppressants that that suppress or reduce the strength of the body’s immune system, like usage of anti-TNF blocking agents, methotrexate, or prednisolone >7.5 mg. ^3^ These included: active intravenous drug use, prosthetic heart valve, previous endocarditis and congenital heart disease. * Fisher’s exact test.

**Table 2 nutrients-12-03131-t002:** Comparison of the clinical characteristics.

Clinical Characteristics	All Cases *n* = 620	TPN Cases *n* = 53	Non-TPN Cases *n* = 567	*p*-Value
MRSA (%)	13 (2.1)	1 (1.9)	12 (2.1)	1.00 *
*Mode of acquisition (%)*				
Community acquired	206 (33.2)	0	206 (36.3)	<0.001
Healthcare associated	195 (31.5)	44 (83)	151 (26.6)	
Hospital acquired	219 (35.3)	9 (17)	210 (37)	
*Portal of entry (%)*				<0.001
Skin	194 (31.6)	3 (5.7)	191 (34)	
Central venous line	103 (16.8)	48 (90.6)	55 (9.8)	
Peripheral venous line	97 (15.8)	1 (1.9)	96 (17.1)	
Lungs	14 (2.3)	0	14 (2.5)	
Other ^1^	36 (5.9)	0	36 (6.4)	
Unknown	170 (27.7)	1 (1.9)	169 (30.1)	
Fever at onset (%)	490 (84.6)	41 (83.7)	449 (84.7)	0.83
Persistent fever >72 h treatment (%)	180 (31.9)	7 (17.5)	173 (33)	0.04
*Follow-up blood cultures performed (%)*	552 (90.8)	45 (95.7)	507 (90.4)	0.22
Positive at 48 h	152 (24.5)	13 (24.5)	139 (24.5)	1.00
CRP at onset mg/l, mean (SD)	161 (127)	74.6 (67.9)	169 (128)	<0.001
Creatinine at onset umol/l, mean (SD)	121 (104)	113 (66.4)	121 (107)	0.58
*^18^FDG-PET/CT scan performed (%)*	334 (55)	37 (72.5)	297 (52.8)	<0.01
Metastatic infection	233 (70.6)	20 (54.1)	213 (72.7)	0.02
Spondylodiscitis	50 (8.1)	1 (1.9)	49 (8.6)	0.11
Endocarditis	34 (5.5)	1 (1.9)	33 (5.8)	0.35
Skin/soft tissue	85 (13.7)	3 (5.7)	82 (14.5)	0.09
Spleen	3 (0.5)	0	3 (0.5)	1.00 *
Liver/bile system	4 (0.6)	0	4 (0.7)	1.00 *
Psoas muscle	18 (2.9)	1 (1.9)	17 (3)	1.00 *
Non-vertebral osteomyelitis	21 (3.4)	1 (1.9)	20 (3.5)	1.00 *
Arthritis	56 (9)	3 (5.7)	53 (9.3)	0.46 *
Brain	4 (0.6)	0	4 (0.7)	1.00 *
Endovascular	54 (8.7)	10 (18.9)	44 (7.8)	<0.01
Lung	61 (9.8)	15 (28.3)	46 (8.1)	<0.001
Other	64 (10.3)	0	64 (11.3)	<0.01 *
Endocarditis ^2^ (%)	63 (10.2)	2 (3.8)	61 (10.8)	0.15 *
ID bedside consultation (%)	473 (78.7)	37 (88.1)	436 (78)	0.12
*Onset treatment ^#^ (%)*				0.98
<24 h	284 (49)	26 (51)	258 (48.8)	
24–48 h	88 (15.2)	7 (13.7)	81 (15.3)	
48 h–72 h	63 (10.9)	5 (9.8)	58 (11)	
>72 h	145 (25)	13 (25.5)	132 (25)	
*Antibiotic therapy (%)*				0.13
Flucloxacillin	523 (85.5)	43 (84.3)	480 (85.6)	
Cefazolin	32 (5.2)	3 (5.9)	29 (5.2)	
Vancomycin/teicoplanin	19 (3.1)	4 (7.8)	15 (2.7)	
Other	38 (6.2)	1 (2.0)	37 (6.6)	
*Duration of therapy (%) ^##^*				0.04
<2 weeks	40 (6.5)	2 (3.8)	38 (6.7)	
2–4 weeks	192 (31.2)	22 (41.5)	170 (30.2)	
4–6 weeks	71 (11.5)	5 (9.4)	66 (11.7)	
>6 weeks	179 (29.1)	16 (30.2)	163 (29)	
Died during therapy	115 (18.7)	4 (7.5)	111 (19.7)	
Switch to oral therapy (%)	191 (32.4)	5 (10.4)	186 (34.4)	<0.001 *
Intensive care admission (%)	173 (28.1)	12 (23.1)	161 (28.6)	0.40
Duration hospital admission, mean (SD) ^###^	21.3 (20.7)	26 23)	20.8 (20.5)	0.05

**Abbreviations:** CRP: C-reactive protein, ^18^F-FDG PET/CT: 18-fluor-FDG positron emission tomography/computed tomography, ID: infectious disease, MRSA: methicillin-resistant *Staphylococcus aureus*, SD: Standard deviation, TPN: total parenteral nutrition. ^1^ Other portals of entry, mostly urinary tract. ^2^ Endocarditis was diagnosed according to Dukes criteria with echocardiography and/or ^18^F-FDG-PET/CT scanning. ^#^ Calculated as time in days between date of onset symptoms and start date of appropriate antibiotic treatment. ^##^ Calculated as time in days between start date of appropriate antibiotic treatment and stop date of antibiotic treatment. ^###^ Calculated as time in days between date of positive blood culture and discharge date. * Fisher’s exact test.

**Table 3 nutrients-12-03131-t003:** In-hospital and cumulative mortality of TPN and non-TPN cases.

	All Cases *n* = 620	TPN Cases *n* = 53	Non-TPN Cases *n* = 551	*p*-Value	Missing Values TPN/Non-TPN
In-hospital death	103 (17.1)	2 (3.8)	101 (18.3)	0.004	0/16
*1 month*					
Death	111 (18.4)	1 (1.9)	110 (20)	<0.001	0/16
*3 months*					
Death	152 (25.2)	5 (9.4)	147 (26.7)	0.004	0/16
Relapse	9 (1.5)	2 (3.8)	7 (1.3)	0.18	0/16
Overall mortality ^a^	181 (31.8)	5 (9.8)	176 (34)	<0.001	2/49
SAB attributable mortality	93 (15)	1 (1.9)	92 (17)	0.002	2/36
Complicated SAB ^b^	246 (39.7)	22 (41.5)	224 (39.5)	0.78	0/0

**Abbreviations:** SAB: *Staphylococcus aureus* bacteremia, TPN: total parenteral nutrition. ^a^ Cumulative outcomes at 6 months. ^b^ Definition of complicated SAB: infective endocarditis, metastatic infection, non-retainable infected foreign body or relapse of infection.

**Table 4 nutrients-12-03131-t004:** Cox proportional hazard model for overall mortality with patient characteristics.

	Univariate Analysis ^a^			Multivariate Analysis		
	HR	95% CI	*p*-Value	HR	95% CI	*p*-Value
Gender (ref.: female)	1.06	0.78–1.43	0.71			
Charlson com. index (per point)	1.28	1.19–1.37	<0.001	1.32	1.24–1.4	<0.001
Immunocompromised (ref.: no)	1.29	0.93–1.78	0.13			
Foreign body material (ref.: no)	1.01	0.75–1.38	0.93			
Risk factors for endocarditis (ref.: no)	1.63	1.08–2.45	0.02	1.74	1.15–2.62	<0.01
TPN support (ref.: no)	0.25	0.10–0.60	<0.01	0.38	0.16–0.93	0.03

**Abbreviations:** HR: hazard ratio, CI: confidence interval, Ref: reference category, TPN: total parenteral nutrition. ^a^ All cases of SAB were included and univariate models were adjusted for TPN support. Predictive values (*p* < 0.1) were included in the multivariate model.

**Table 5 nutrients-12-03131-t005:** Comparison of TPN patients with and without complicated *S. aureus* bacteremia.

	Total TPN Cases *n* = 53 (%)	Complicated SAB *n* = 22 (%)	Non-Complicated SAB *n* = 31 (%)	*p*-Value	Missing Values
Healthcare-associated onset	44 (83)	21 (95)	23 (74)	0.04	0
Fever at onset	41 (77)	17 (77)	24 (77)	0.27 *	4
Inflammation at CVC insertion	25 (47)	12 (55)	13 (42)	0.31	15
Previous problems with CVC ^#^	12 (22)	9 (41)	3 (10)	0.02 *	15
CRP, mean (SD)	75 (68)	99 (66)	54 (64)	0.02	5
Therapy onset <24 h ^##^	26 (49)	4 (18)	22 (71)	<0.001 *	2
CVC removed	51 (96)	21 (95)	30 (97)	0.24	1
Persistent fever >72 h	7 (13)	4 (18)	3 (10)	0.69 *	13
Positive follow-up BC	13 (25)	10 (45)	3 (10)	<0.01 *	0
SAB in medical history	14 (26)	5 (23)	9 (29)	0.51	12

**Abbreviations:** BC: blood culture; CVC: central venous catheter; CRP: C-reactive protein; SAB: *S. aureus* bacteremia; TPN: total parenteral nutrition. ^##^ Previous problems with CVC included previous tunnel or exit site infection, dislocation and catheter-related thrombosis. ^#^ Calculated as time in days between date of onset symptoms and start date of appropriate antibiotic treatment. * Fisher’s exact test.

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
