# Peer review of "Clinical Characteristics and Outcomes of S. Aureus Bacteremia in Patients Receiving Total Parenteral Nutrition"

_nutrients, 2020, doi:10.3390/nu12103131_

Round 1
Reviewer 1 Report
Gompelman and colleagues present a straightforward, retrospective epidemiological overview involving observational studies of SAB (Staphylococcus aureus bacteremia) patients over a seven-year period, comparing those that are receiving Total Parenteral Nutrition (TPN) and to non-TPN cases. Long term TPN is known to be correlated to higher incidence of SAB, due to the insertion of a catheter (peripherally or centrally). They describe specific clinical characteristics observed in the TPN-related SAB, specifying that the study group more frequently had younger, female patients, with lower mortality rates than non-TPN-related SAB cases. Generally, their results agreed with other clinical studies focusing on patients with a Central Venous Catheter (CVC), which would be important given the other limitations of the study. A main suggestion though to improve this paper, is to also provide an insight into the current treatment protocol in place for TPN patients in the Netherlands, as what the main conclusion of the study focuses on was the lower mortality rates observed in TPN patients who present early to the hospital compared to the non-TPN patients. A look into the patient care management may provide more value to the findings of this research.
Other questions and suggestions are as follows:
ABSTRACT
Line 15: Change "Especially" to "Specifically". Also italicize Staphylococcus aureus (also in Line 56 and 130)
MATERIALS AND METHODS
Line 58-59: Is this selection inclusive or irrelevant of polymicrobial infections? If excluded, please indicate in the study design.
Line 76: infectious disease clinic?
Line 78: Correct to "cleared from the requirement of approval by an ethics committee". However, please clarify the nature of the data after 2016 which now required approval.
Line 92: Change "according" -> "according to" (also in line 157, 278)
Line 102-104: Would be informative to see this data in the patient characteristics overview (Table 1).
Line 107-108: Describe in the statistical analysis method the consideration for choosing the chi-square or the Fisher exact test for either categorical variable since they are both showing up in the figures.
Line 110-111: How was the data between patients that were treated in hospitals vs. those receiving home parenteral nutrition treated?
RESULTS
Table 4. Table must be fixed to match labels with values.
Figure 3 and 4. Please improve quality of these figures.
DISCUSSION
Line 224-226: I believe an important discussion to include here is the common treatment regimen practiced by the hospitals (antibiotic treatment, etc.). Could age also be a factor in the comparison as the TPN patients were younger than the non-TPN patients?
Line 234: change "a" to "an"
Line 253: change "Thus, only in 4 patients the metastatic infection was diagnosed" to "Thus, only in 4 patients was the metastatic infection diagnosed.."
Line 266-267: please indicate the in-citation reference for Fowler et al. and Zakhem et al. (also similarly in other parts of the discussion. describe which reference study was referred to).
Reviewer 2 Report
This manuscript by M Gompelman and colleagues described the characteristics, clinical course, and outcome of SAB in patients with and without TPN support. They found that patients on TPN have a milder course of SAB with lower mortality rates compared to non-TPN SAB patients. The study is well-designed, executed and the results are convincing. This may be helpful to develop new strategies for preventing or treating severe bloodstream infections.
Author Response
Thank you.
Action taken: none